# Neuronal integration in the adult mouse olfactory bulb is a non-selective addition process

**Jean-Claude Platel[1]\*, Alexandra Angelova[1], Stephane Bugeon[1], Jenelle Wallace[2], Thibault Ganay[1], Ilona Chudotvorova[1], Jean-Christophe Deloulme[3], Christophe Béclin[1], Marie-Catherine Tiveron[1], Nathalie Coré[1], Venkatesh N Murthy[2], Harold Cremer[1]**

[1]Aix-Marseille University, CNRS, IBDM, UMR 7288, Marseille, France; [2]Department of Molecular & Cellular Biology, Harvard University, Cambridge, United States; [3]Grenoble Institut des Neurosciences, Université Grenoble Alpes, Grenoble, France

**Abstract** Adult neurogenesis in the olfactory bulb (OB) is considered as a competition in which neurons scramble during a critical selection period for integration and survival. Moreover, newborn neurons are thought to replace pre-existing ones that die. Despite indirect evidence supporting this model, systematic in vivo observations are still scarce. We used two-photon in vivo imaging to study neuronal integration and survival. We show that loss of new neurons in the OB after arrival at terminal positions occurs only at low levels. Moreover, long-term observations showed that no substantial cell death occurred at later stages. Neuronal death was induced by standard doses of thymidine analogs, but disappeared when low doses were used. Finally, we demonstrate that the OB grows throughout life. This shows that neuronal selection during OB-neurogenesis does not occur after neurons reached stable positions. Moreover, this suggests that OB neurogenesis does not represent neuronal turnover but lifelong neuronal addition.
DOI: https://doi.org/10.7554/eLife.44830.001

\*For correspondence:
jean-claude.platel@univ-amu.fr

**Competing interests:** The authors declare that no competing interests exist.

## Introduction

Neurogenesis continues after birth in the hippocampus and olfactory bulb of rodents. During OB neurogenesis predetermined stem cell population along the walls of forebrain ventricles generate neuronal precursors that migrate via the rostral migratory stream (RMS) into the center of the OB. After their radial migration into the principal target layers, the granule cell (GCL) and glomerular layers (GL), cells integrate into the preexisting circuitry and function as interneurons using GABA and dopamine as their principal neurotransmitters (*Whitman and Greer, 2007*).

The currently available information indicates that OB neurogenesis is based on two key principles: First, neuronal integration in the adult is a competitive process, during which large numbers of newly arriving neurons compete for integration into the circuitry and ultimately survival. This competition is thought to occur during a defined critical window of 2–8 weeks after arrival and leads to the apoptotic elimination of about half of the initial population (*Bergami and Berninger, 2012*; *Lledo et al., 2006*; *Mandairon et al., 2006*; *Petreanu and Alvarez-Buylla, 2002*; *Winner et al., 2002*; *Yamaguchi and Mori, 2005*). Second, the OB represents a turnover system, in which newly integrating cells replace preexisting ones, leading to a relatively stable total number of neurons in the target layers (*Bergami and Berninger, 2012*; *Imayoshi et al., 2008*; *Lledo et al., 2006*).

These two concepts are to a large extend based on lineage-tracing experiments using thymidine analogs like bromodeoxyuridine (BrdU) or 3H-thymidine to label the DNA of dividing cells (*Mandairon et al., 2006*; *Petreanu and Alvarez-Buylla, 2002*; *Winner et al., 2002*; *Yamaguchi and*

*Mori, 2005*). A common observation in such experiments is a loss of labeled cells during the first few weeks after their arrival in the olfactory bulb, which led to the postulation of a selection mechanism allowing the remodeling of specific OB circuits during a period when new cells had already matured and developed dendritic arborizations (*Petreanu and Alvarez-Buylla, 2002*).

Alternatively, genetic approaches using CRE-inducible markers layers have been performed and demonstrated an accumulation of adult born neurons in the OB over time (*Imayoshi et al., 2008*). In agreement with the turnover model, this has been interpreted as a replacement of older neurons that died (*Imayoshi et al., 2008*). Only recently more direct approaches based on two-photon in vivo imaging allowed studying OB neurons directly in the living animal (*Mizrahi et al., 2006*; *Sailor et al., 2016*; *Wallace et al., 2017*). Interestingly, long-term observation of either juxtaglomerular neurons in general (*Mizrahi et al., 2006*), or more specifically of dopaminergic neurons, demonstrated an increase in these populations over time (*Adam and Mizrahi, 2011*). While at first sight this finding contradicts a pure replacement model, it was interpreted as a change in the interneuron subtype composition of the OB (*Adam and Mizrahi, 2011*).

In addition, olfactory activity and learning have been implicated in the regulation of neuron survival. On one hand, sensory deprivation by naris closure reduced the number of BrdU-labeled newborn neurons in the OB (*Mandairon et al., 2006*; *Saghatelyan et al., 2005*; *Yamaguchi and Mori, 2005*). On the other hand, olfactory training increased the number of labeled neurons (*Mouret et al., 2008*).

Thus, while the available data is still mostly indirect, the elegant model based of selection and replacement appears justified. However, to doubtlessly validate this model and to understand the factors controlling the adult neurogenic process, all populations of integrating neurons have to be observed in the living animal from their arrival in the OB throughout the selection phase until their disappearance.

Here, we combined genetic birthdating and lineage tracing with long term in vivo microscopy to follow timed cohorts of postnatal and adult born neurons from their arrival in the OB for up to 6 months. Quantitative analyses demonstrate that neuronal loss during the critical period for survival, and also at later stages, is rare in all observed populations. We demonstrate that classically used doses of the tracers BrdU and 5-ethynyl-2'-deoxyuridine (EdU) induce cell loss. This loss was not observed when low doses of EdU were used. Finally, based on in vivo microscopy and light-sheet imaging of fixed cleared tissue, we show that neuronal addition merely than replacement occurs in the adult OB, leading to permanent growth of the structure.

## Results

### Long-term in vivo imaging of postnatal and adult born OB neurons

We used two-photon imaging to directly study the integration and survival of perinatal and adult born OB neurons at high spatial and temporal resolution in the living animal. We first focused on the perinatal period, when most OB interneurons are generated (*Batista-Brito et al., 2008*). Postnatal in vivo brain electroporation of the dorsal ventricular zone targets stem cell populations that generate neurons for the superficial layers of the OB (*Figure 1a,b*; *de Chevigny et al., 2012b*), which can be reliably reached by two-photon microscopy (*Adam and Mizrahi, 2011*). We used this dorsal targeting approach to introduce a CRE-expression plasmid into R26-RFP reporter mice (*Figure 1a*). Three weeks later, OB-labeled neurons comprised a mixed population of 6% tyrosine hydroxylase expressing dopaminergic/GABAergic neurons, 12% calretinin positive purely GABAergic neurons, 22% other GABAergic PGN (*Figure 1b*) and 60% mostly superficially positioned granule cells (GC).

An adaptation of the reinforced thin skull method allowed for frequent and long-term imaging of awake mice while perturbing the physiology of the OB only minimally (*Drew et al., 2010*). In agreement with previous observations (*Xu et al., 2007*), there was no detectable astroglia reaction or accumulation of microglia after thinning and window implantation (*Figure 1—figure supplement 1*).

Three weeks after electroporation, when skull growth was sufficiently advanced, thin-skull preparation was performed and populations of individually identified neurons in the glomerular layer (GL; *Figure 1d–g*) and the granule cell layer (GCL, *Figure 1—figure supplement 2*) were imaged in awake animals at high resolution over the following weeks and months. All analyzed neurons were individually identified in Z-stacks (*Video 1*) based on relative position and morphology. Neurons

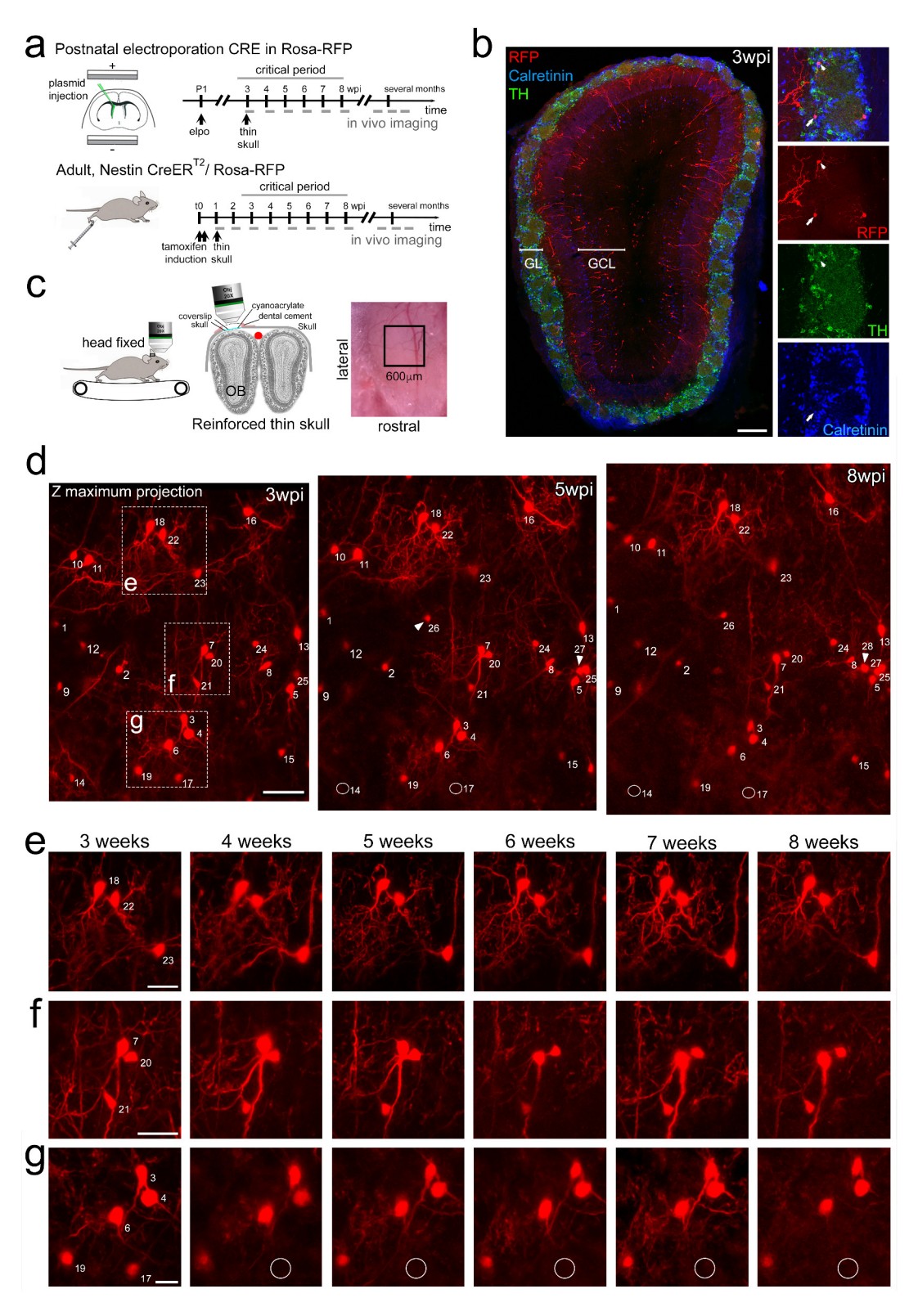

**Figure 1.** Long-term in vivo imaging in the OB. (**a**) Induction protocols and imaging timeline. In perinatal animals, a CRE-expression plasmid was introduced in the dorsal neural stem cell compartment of Rosa-RFP mice using postnatal electroporation. To label neurons in adults, Nestin-CreERT2 animals were bred to Rosa-RFP mice and induced with tamoxifen at 2 months of age. Thin skull preparation was routinely performed one-week post induction. A weekly imaging scheme was implemented over the critical period and up to 5 months. (**b**) Postnatal in vivo brain electroporation at P1-P4

*Figure 1 continued on next page*

*Figure 1 continued*

leads at three wpi to the appearance of various interneuron types, including TH and CR expressing subtypes, in the superficial GCL and the GL layers of the OB. (c) In vivo microscopy setup. Mice were imaged with the head fixed to the two-photon microscope. Animals could move on a treadmill but rarely did so during imaging sessions. Thin skull preparation allowed high-resolution imaging on a weekly basis. (d) Example of an image Z-stack showing 25 individually identified neurons from 3, 5, and 8 weeks after CRE electroporation. Note that neurons 14 and 17 are lost (circles) while several neurons are added (arrowheads). (e, f, g) High-resolution images of weekly observations of three groups of neurons highlighted in d. Cell substructures, dendrites and minor cell displacements can be followed over time. Scale bar: 200 µm in b, 50 µm in d, 30 µm in e, f, g.

DOI: https://doi.org/10.7554/eLife.44830.002

The following figure supplements are available for figure 1:

**Figure supplement 1.** Absence of inflammation after thin skull surgery.
DOI: https://doi.org/10.7554/eLife.44830.003
**Figure supplement 2.** Long term imaging of postnatally born granule neurons.
DOI: https://doi.org/10.7554/eLife.44830.004
**Figure supplement 3.** Image and table exemplifying how perinatally (a,b) and adult (c,d) generated neurons were scored over the critical period (for a, b compare also *Figure 1d*).
DOI: https://doi.org/10.7554/eLife.44830.005

were numbered and revisited weekly over the next months (*Figure 1d–f*; *Video 1*; *Figure 1—figure supplement 3a,b*). After identification of the first cohort, smaller numbers of additional neurons appeared permanently in the observation window as a consequence of ongoing neurogenesis (arrowheads in *Figure 1d*, 5wpi). These were also numbered and followed and used for long-term analyses (Figure 6). Neurons in the observation field showed stable relative positions over time (*Figure 1d*); however, in some cases, minor positional adjustment were observed that could be followed over subsequent imaging sessions (*Figure 1d,f*). Generally, resolution was sufficient to observe even minor changes in dendritic organization of neurons over time (see neuron no. 7 in *Figure 1d,f*).

## In vivo observation of perinatally born neurons

Based on this direct and systematic imaging approach, we first focused on perinatally born neurons survival during the proposed critical selection period, thus until 8 weeks after their generation at the ventricles (*Mandairon et al., 2006*). Neurons that were present during the first observation time point (3 weeks after electroporation of the respective stem cells) were followed over the next 5 weeks. Among 755 periglomerular neurons (PGN) in 11 mice only 5.1% were lost over the proposed critical period (*Figure 2a*, see circles for lost cells no. 14 and 17 in *Figure 1d, g*). The percentage of lost neurons was very similar between individual animals and was independent of the density of labeled cells in the observation window (between 18 and 100 neurons; *Figure 2a*).

Next, we investigated newborn granule cells (GCs) in the underlying GCL in six mice with particularly high-quality and stable window preparations (*Figure 1—figure supplement 2*). Out of 178 RFP positive neurons observed between 3 and 8 weeks after their birth not a single cell disappeared over the subsequent imaging sessions (*Figure 2b*). We conclude that perinatally generated OB interneurons in both, the GL and the GCL are rarely eliminated after arrival in terminal positions.

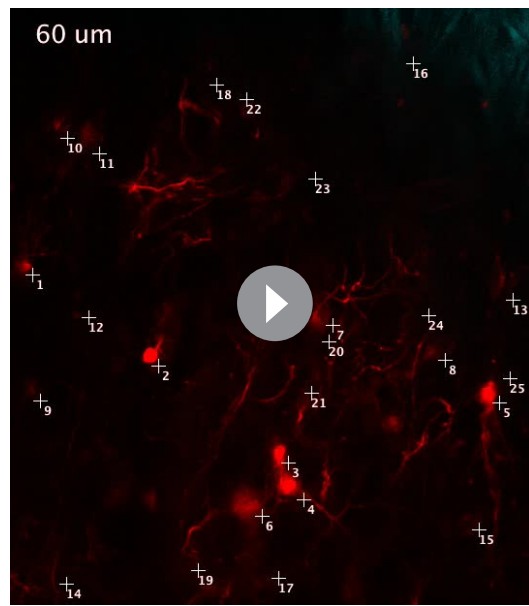

**Video 1.** Example of a Z-stack showing perinatally born neurons in the GL. This stack was the basis for the projection presented in *Figure 1d*.
DOI: https://doi.org/10.7554/eLife.44830.006

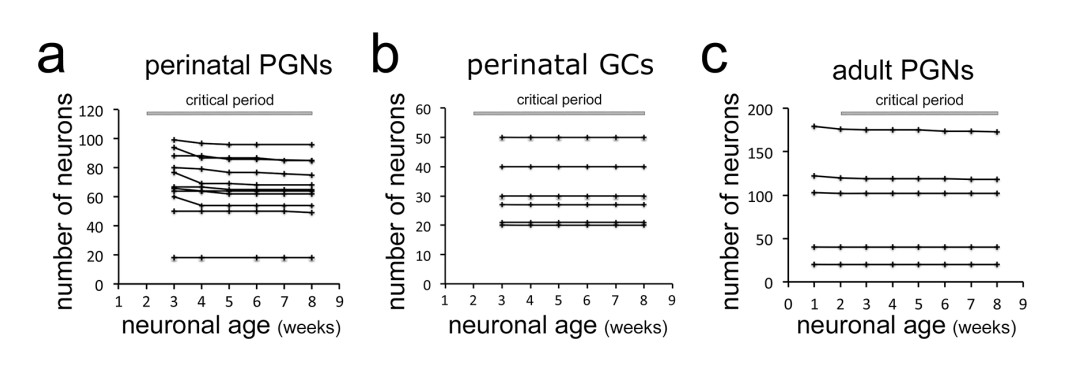

**Figure 2.** Stability of neuron populations in the OB in vivo. (**a**) Tracing of perinatally induced timed neuron first cohorts (755 neurons) in 11 mice from 3 to 8 wpi. (**b**) Tracing of perinatally induced first cohorts of granule cells (178 cells in six mice) during the proposed critical period. (**c**) Tracing of the first cohort of periglomerular neurons in eight adult animals (538 neurons) after induction with tamoxifen injection at 2 months.
DOI: https://doi.org/10.7554/eLife.44830.007
The following figure supplement is available for figure 2:

**Figure supplement 1.** Adult born neuron labeling using Nestin-Cre-ERT2/Rosa -RFP mice.
DOI: https://doi.org/10.7554/eLife.44830.008

## In vivo observation of adult born neurons

We then investigated the stability of adult born neurons during the critical selection period. First, we focused on PGN that can be reliably imaged after thin skull preparation. As in vivo electroporation is inefficient in adult mice, we crossed the Rosa-RFP line with Nestin Cre-ER[T2] mice (*Lagace et al., 2007*) and induced a heterogeneous cohort of labeled newborn neurons by tamoxifen injection at 2 months of age (*Ninkovic et al., 2007*) (*Figure 2—figure supplement 1a,b*). One week after induction virtually all RFP-positive cells in the RMS and about 30% in the OB layers expressed the immature neuron marker doublecortin (*Figure 2—figure supplement 1c,d*). Reinforced thin skull preparation was routinely performed at 1 week post-induction (wpi). Weekly observations of individually identified PGN in the GL were performed as described above (*Video 2*, *Figure 1—figure supplement 3c,d*). Analyses of 538 periglomerular neurons of the first cohort in eight animals showed that only 1.5% disappeared over the 7 weeks period after their first identification (*Figure 2c*).

Finally, we investigated the survival of adult born granule neurons after their arrival in the OB. To access this deeply positioned and densely packed cell population, we used a cranial window preparation in Nestin Cre-ER[T2]/Rosa-RFP mice. We observed 101 adult born neurons in nine animals (*Figure 3a,b,c*). During the 8 weeks observation window, six neurons disappeared (5,9%). In addition, we labeled adult born granule cells with an injection of a tomato-expressing lentivirus into the RMS (*Figure 3d*; *Wallace et al., 2017*). Tracing of 48 adult-born GCs in 19 fields of view from three mice led to the identification of only a single cell that disappeared (2.1%; *Figure 3e*) during the 7 weeks observation period. Thus, overall under physiological conditions cell loss in adult born OB neurons during the proposed critical

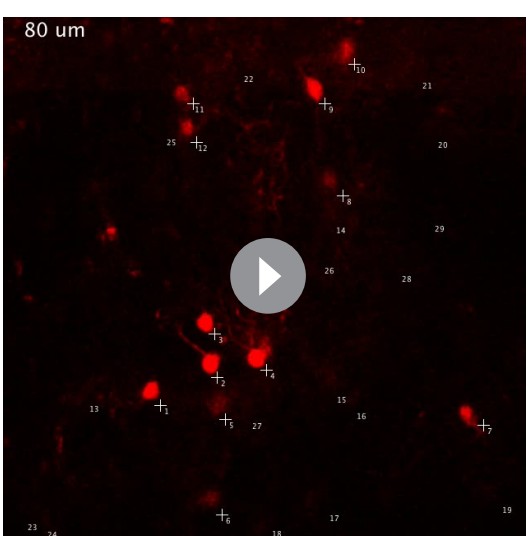

**Video 2.** Example of a Z-stack showing adult born neurons in the GL. This stack was the basis for the projection presented in Extended Data *Figure 1—figure supplement 3*.
DOI: https://doi.org/10.7554/eLife.44830.009

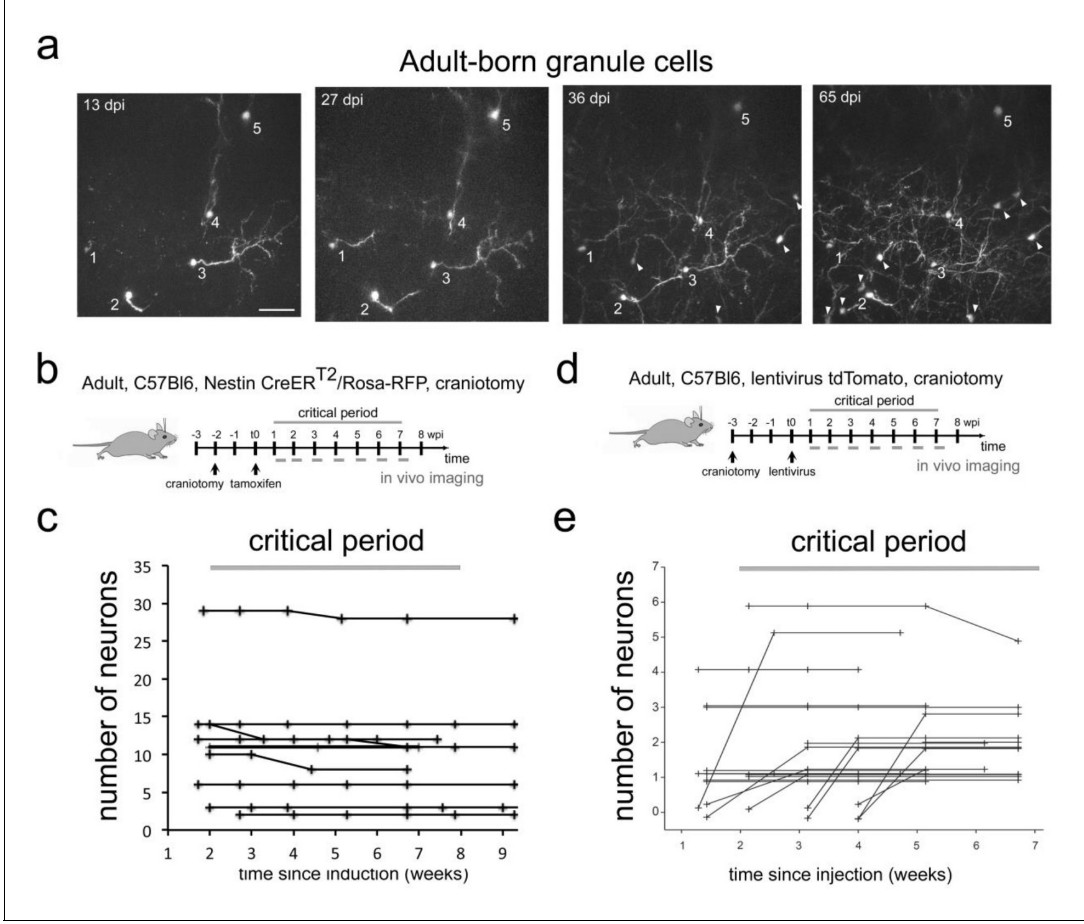

**Figure 3.** Adult-born granule cells stability in the OB in vivo. (a) Example field of view (maximum intensity projection of volume with depth 150 μm) showing cells that were tracked between 13 and 65 days post tamoxifen injection (dpi). Arrowheads indicate newcomers that appeared after the first day of imaging and were subsequently stable at later time point. Scale bar 50 μm. (b) Labeling protocols and imaging timeline. To label granule neurons in adults, Nestin CRE-ER$^{T2}$/Rosa-RFP animals were induced with tamoxifen. A craniotomy was performed 2 weeks before tamoxifen injection. A weekly imaging scheme was implemented over the critical period. (c) Tracing of adult-born granule cells (101 neurons) in nine mice over the critical period. (d) Labeling protocols and imaging timeline. To label granule neurons in adults, C57Bl6 animals were injected with a tomato lentivirus in the RMS. A craniotomy was performed 3 weeks before lentivirus injection. A weekly imaging scheme was implemented over the critical period. (e) Tracing of adult-born granule cells (48 neurons) in 3 mice from 19 fields of view from 1 to 7 weeks post injection in the RMS. Lines beginning at zero indicate new cells that appeared in the field of view and were subsequently tracked.

DOI: https://doi.org/10.7554/eLife.44830.010

selection period was very low, comparable to the findings for perinatally born neurons.

Next we asked if neuron loss could be detected in non-physiological situations. It has been shown that olfactory sensory deprivation induces cell death in adult born OB neurons (*Mandairon et al., 2006*; *Saghatelyan et al., 2005*; *Yamaguchi and Mori, 2005*). To investigate if increased cell death could be observed in our imaging paradigm, we performed naris closure in adult Nestin Cre-ER$^{T2}$/Rosa-RFP mice 1 week after tamoxifen induction (*Figure 4a*). Analysis of RFP-positive PGN over the following 8 weeks revealed a significant increase in cell loss (*Figure 4bc*; p=0.03, four control and three occluded animals, 151 cells).

In conclusion, under physiological conditions newly born neurons in the perinatal and adult OB show little cell loss after arrival in the OB. However, significant cell loss during this observation period was found after blocking sensory input, demonstrating that cell death could be detected with our approach.

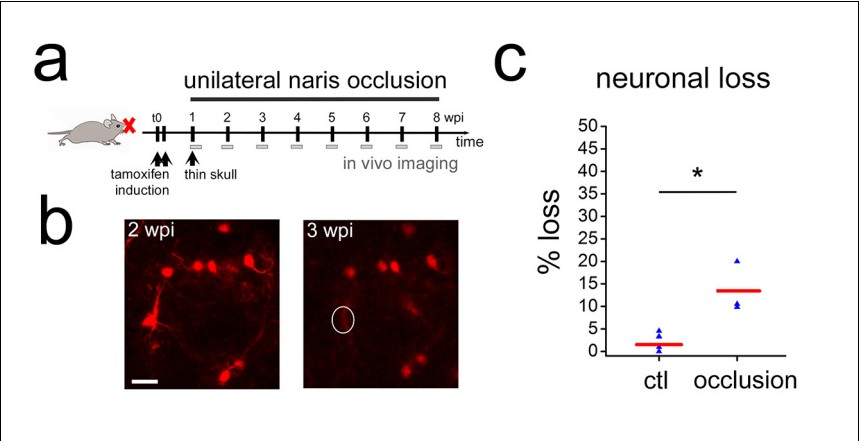

**Figure 4.** Sensory deprivation in the OB lead to neuronal death. (a) Timeline for sensory deprivation experiment. Naris occlusion and thin skull preparation were performed 1 week after induction of RFP-positive neurons in adult mice. (b) Two weeks after occlusion neurons with complex morphologies were lost in the OB. (c) Quantification of neuron loss in control and occluded OBs over 8 weeks. Scale bar: 20 μm in b.
DOI: https://doi.org/10.7554/eLife.44830.011

## Dose dependent toxicity of thymidine analogs

The above findings were at odds with the existence of a critical period for survival during which, under normal conditions, about half of the adult born neurons are removed from the OB by cell death (*Mandairon et al., 2006*; *Mouret et al., 2008*; *Petreanu and Alvarez-Buylla, 2002*; *Winner et al., 2002*; *Yamaguchi and Mori, 2005*). This concept is to a large extent based on tracing of timed cohorts of newborn neurons using the integration of thymidine analogs, most often BrdU, into the DNA of dividing cells. To investigate if these differences were due to our particular experimental conditions we first repeated such pulse chase studies using commonly used doses of BrdU and following established protocols (*Mandairon et al., 2006*; *Mouret et al., 2008*; *Whitman and Greer, 2007*). Indeed, using four i.p. injections of 50 mg/kg BrdU every 2 hr into adult mice, we found an approximately 40% loss of labeled neurons in the OB between 2 and 6 weeks in the GL as well as in the GCL (*Figure 5ab*). As in our direct imaging approach, we focused on the dorsal aspect of the OB, we investigated if in this region BrdU-positive cells showed a different behavior than in the rest of the structure. BrdU+ cell number in the dorsal OB showed the same 40% loss that was found in the entire bulb (*Figure 5b*).

Altogether, these DNA-labeling based findings were in full agreement with previous studies, showing a strong reduction of newborn cells during the critical selection period (*Mandairon et al., 2006*; *Mouret et al., 2008*; *Petreanu and Alvarez-Buylla, 2002*; *Whitman and Greer, 2007*; *Winner et al., 2002*; *Yamaguchi and Mori, 2005*). However, they strongly contradicted our in vivo observations showing very little cell loss. As suggested before (*Lehner et al., 2011*), we considered the possibility that incorporation of modified nucleotides impacted on neuronal survival in the OB and developed an approach to test this hypothesis.

To allow immunohistological BrdU detection, tissue samples have to be subjected to strong denaturing conditions that break the complementary base-pairing of DNA, a prerequisite for efficient BrdU antibody binding. Such treatment invariably leads to sample degradation and negatively impacts on staining intensity (*Salic and Mitchison, 2008*). In concert with limitations due to antibody penetration, this imposes the use of relatively high concentrations of BrdU, generally several injections of 50 to 200 mg/kg of body weight i.p., for reliable detection (*Brown et al., 2003*; *Mandairon et al., 2006*; *Mouret et al., 2008*; *Petreanu and Alvarez-Buylla, 2002*; *Whitman and Greer, 2007*; *Winner et al., 2002*). In contrast, EdU labeling of DNA is based on a click-reaction with fluorescent azides (*Rostovtsev et al., 2002*; *Salic and Mitchison, 2008*; *Tornøe et al., 2002*) that have much higher diffusion rates in tissue than antibodies. Moreover, the staining reaction can be can be repeated several times to increase signal strengths and DNA denaturation is dispensable, altogether allowing the use of considerably lower concentrations of EdU in comparison to BrdU

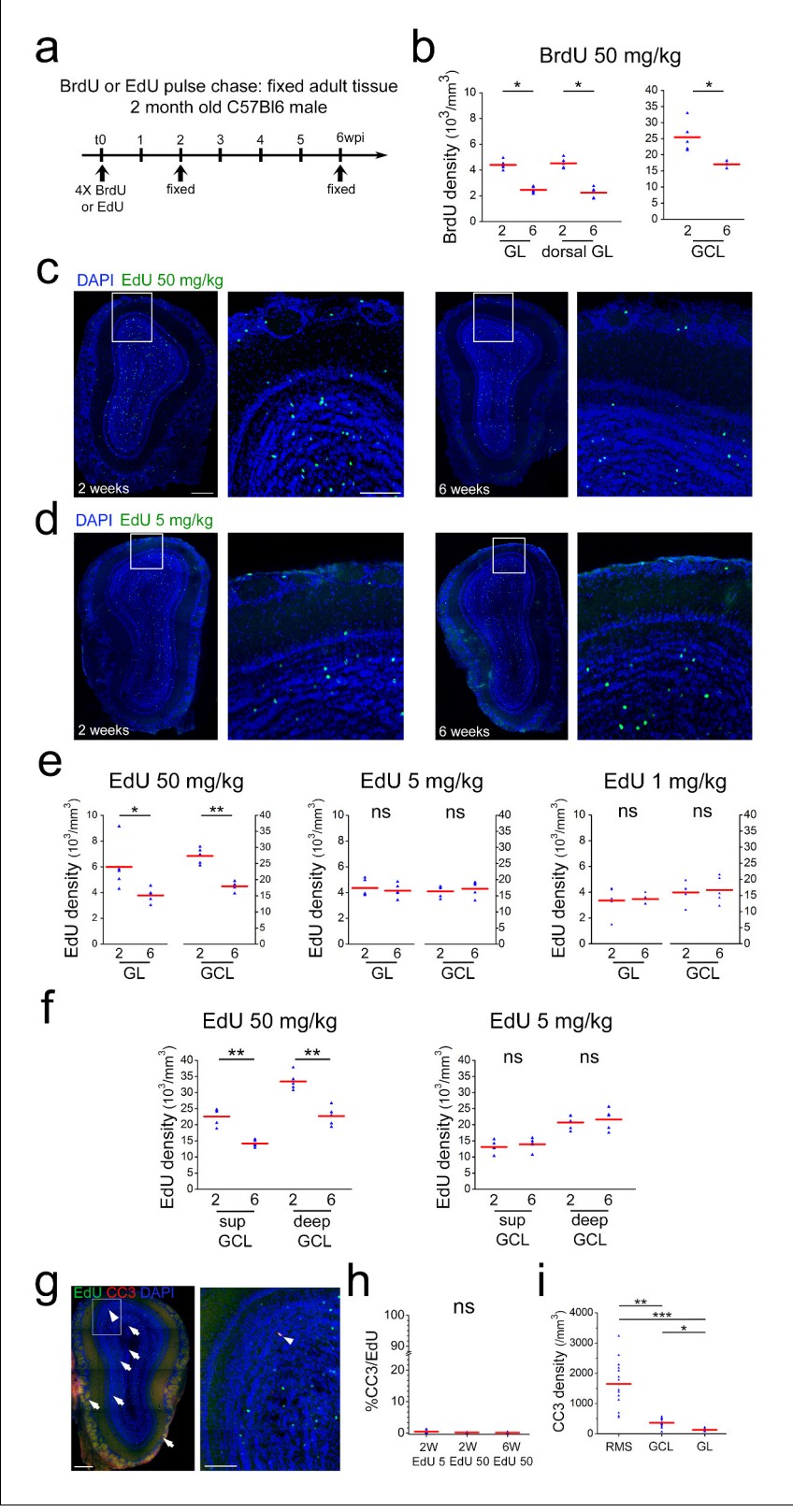

**Figure 5.** Neuronal survival after thymidine analog pulse chase. (a) Timeline of BrdU or EdU pulse chase experiment. Four BrdU/EdU injections at different concentrations were performed in adult mice. Animals were sacrificed 2 and 6 wpi, respectively. (b) BrdU cell density in GL, dorsal GL and GCL between 2 and 6 weeks after injection. (c) Example of immunohistochemical staining of a coronal OB section 2 and 6 weeks after 4 × 50 mg/kg

*Figure 5 continued on next page*

*Figure 5 continued*

EdU injection. (d) Example of immunohistochemical staining of coronal OB sections 2 and 6 weeks after 4 × 5 mg/kg EdU injection. (e) EdU cell density in GL and GCL between 2 and 6 weeks after 4 × 50 mg/kg edU, 4 × 5 mg/kg edU and 4 × 1 mg/kg EdU injection, respectively. Note the strong cell loss at 50 mg/kg of EdU between 2 and 6 weeks and the absence of cell loss at 4 × 5 and 4 × 1 mg/kg EdU. (f) EdU cell density in superficial and deep GCL between 2 and 6 weeks after 4 × 50 mg/kg edU and 4 × 5 mg/kg edU injection, respectively. Note that for EdU 50 kg both layers show a similar cell loss between 2 and 6 weeks while this cell loss is absent in both layers at 5 mg/kg. (g) Example of immunohistochemical staining of a coronal OB section for EdU at 50 mg/kg (green) and cleaved Caspase 3 (red) (h) Increased concentrations of EdU do not augment the number of EdU/cleaved Caspase3 co-labeled cells. (i) Cleaved Caspase three density is more than four times higher in the RMS that in the GCL or GL layers. Scale bar: 300 μm in left panel in c, 100 μm in middle panel in c, 300 and 100 μm in g.

DOI: https://doi.org/10.7554/eLife.44830.012

(*Salic and Mitchison, 2008*). Based on this increased sensitivity, we asked if the concentration of altered nucleotides in the DNA impacts on cell survival of new neurons in the OB.

Four injections of 50 mg/kg EdU in 2-month-old mice led to an about 40% loss of labeled cells in the GL and the GCL of the OB between 2 and 6 weeks, highly comparable to the results based on BrdU (*Figure 5c,e*). Four injection of 1 or 5 mg/kg EdU under the same conditions led to the detection of slightly lower amounts of newly generated cells in the OB layers after 2 weeks (*Figure 5de*). Importantly, under these conditions the decrease in cell numbers between 2 and 6 weeks was not detectable anymore, both in the GL and the GCL (*Figure 5e*). As it has been suggested that deep and superficial GCs are differentially susceptible to replacement (*Imayoshi et al., 2008*), we investigated the impact of low and high doses of EdU on both sub-layers. However, deep and superficial GCs showed the same behavior, cell loss at high EdU dose and survival at 5 mg/kg, as the entire GCL (*Figure 5f*). Altogether, these results lead to the conclusion that loss of labeled cells in the OB during the critical period is correlated with the concentration of modified nucleotides in the DNA of newborn OB neurons.

Programmed cell death has been suggested to underlie the removal of integrating neurons from the OB during the critical window (*Yokoyama et al., 2011*). We investigated Caspase 3 (CC3) expression in the presence of high and low doses of EdU. Analyses of immunostained tissues detected consistently low numbers of CC3 positive cells in the OB layers (363 ± 34 cells /mm$^3$ n = 15; thus about 940 CC3+ cells/GCL) in overall agreement with previous work (770 cells/GCL; *Yamaguchi and Mori, 2005*). However, among the 10803 analyzed EdU-positive OB neurons only 26 showed co-labeling for CC3. Application of four times 5 mg/kg or 50 mg/kg EdU had no influence on the percentage of double stained cells, indicating that thymidine analog induced toxicity did not pass via the apoptotic pathway (*Figure 5h*).

Next, we investigated the distribution of total CC3-positive cells in the OB layers and the RMS. Interestingly, density of CC3-positive cells was more than six times higher in the RMS than in the GCL and the GL (*Figure 5i*). This is in good agreement with previous data (*Biebl et al., 2000*) and suggests that cell death in the system occurs predominantly at the precursor level.

In conclusion, the above results, showing that lineage tracing by high doses of thymidine analogs is associated with cell loss in the OB, point to toxicity of such DNA modifying agents. Moreover, the finding that at low EdU doses cell loss in the OB during the proposed critical selection period is not-detected represents an independent confirmation of our in vivo imaging based findings.

## Neuronal addition in the OB

Neurogenesis in the OB is considered to be a turnover system in which new neurons replace older ones, leading to a relatively stable size of the structure (*Imayoshi et al., 2008*; *Petreanu and Alvarez-Buylla, 2002*). In such a scenario cell loss has to be expected. As we did not observe considerable cell death during early stages in the OB, we asked if neurons disappear at later stages. Continuous long-term observations of perinatally generated PGN and GC provided no evidence for sustained cell loss after the initial 8 weeks time window (*Figure 6a,b*). The same stability of the labeled population was evident when adult generated GC or PGN were observed for up to 24 weeks after their generation (*Figure 6c,d*). Moreover, as CRE-induced recombination in Nestin-CRE-ERT2 mice occurs often at the stem cell level (*Imayoshi et al., 2008*), recombined stem cells continued to

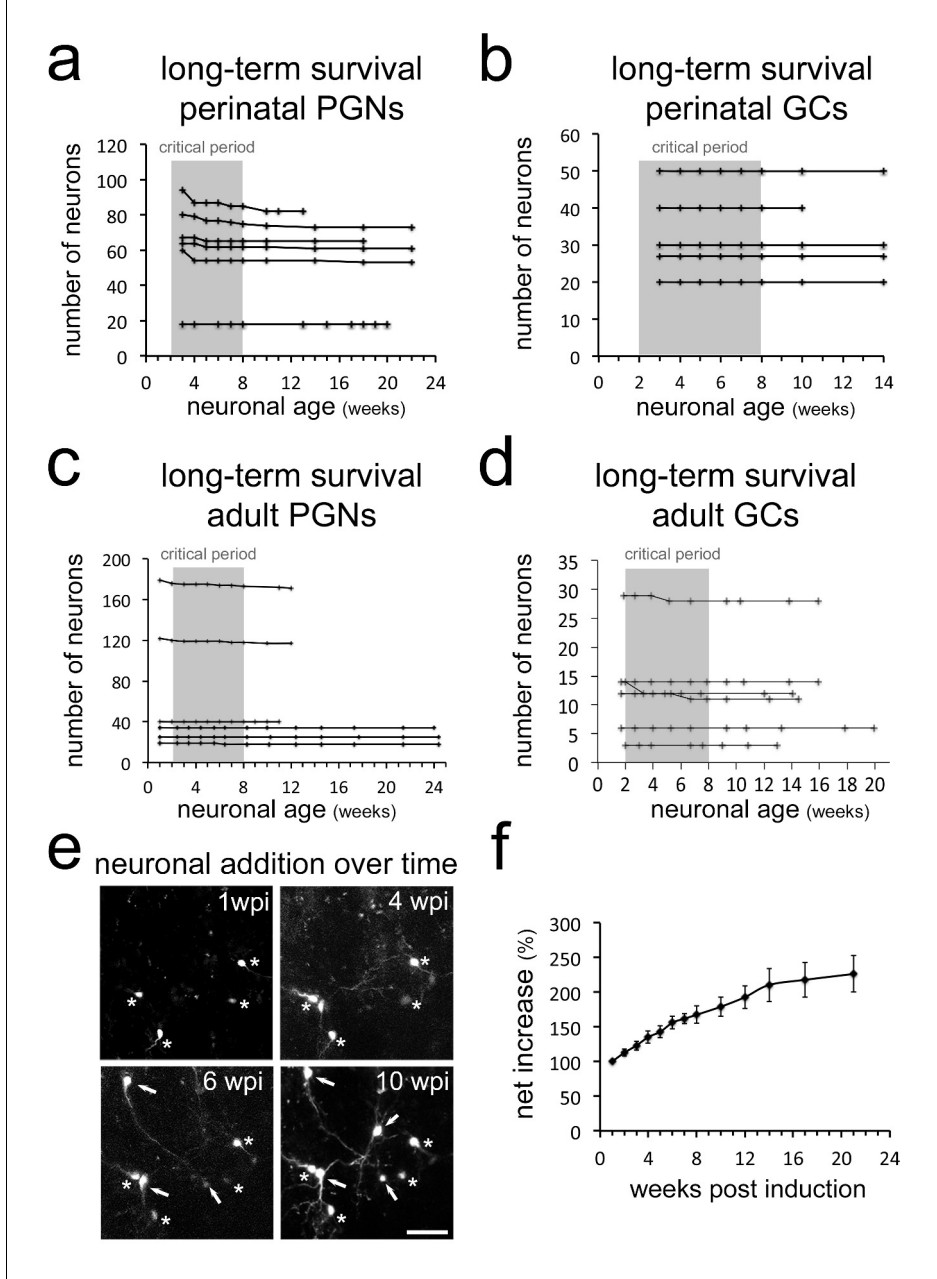

**Figure 6.** Long-term survival of neuron populations and neuronal addition in vivo. Long-term survival of neuron populations and neuronal addition in vivo Very limited cell loss in (**a**) perinatal PGNs, (**b**) perinatal GCs, (**c**) adult born PGNs and (**d**) adult born GCs in long-term in vivo observations. (**e**) Additional RFP expressing neurons appear constantly in the observation window. (**f**) Net increase in all newborn periglomerular neurons in adult Nestin Cre-ERT2 mice over time. Scale bar in 40 μm in e.

DOI: https://doi.org/10.7554/eLife.44830.013

generate new neurons. In agreement, additional adult born neurons permanently appeared in the observation window (*Figure 6e*), leading to a more than doubling of the neuron population of adult generated PGN over an observation period of six months (*Figure 6f*).

How does the OB deal with this permanent addition of neurons in the absence of considerable cell loss? Two potential consequences can be imagined: either the OB grows in size or the cell density in the different layers increases over time. Currently, information about these parameters in the adult rodent OB is based on measurements of serial sections and the available data are in part

contradictory (*Hinds and McNelly, 1977*; *Imayoshi et al., 2008*; *Mirich et al., 2002*; *Petreanu and Alvarez-Buylla, 2002*; *Pomeroy et al., 1990*; *Richard et al., 2010*).

First, we asked if a volume increase in the OB could be detected directly in the living brain during in vivo imaging experiments. We found that over time slightly larger image frames were necessary to accommodate the same group of neurons in our Z-maximum projections of the GCL and GL (*Figure 1d*; *Figure 3a*; *Figure 1—figure supplement 2b*). Using our systematic imaging approach, we quantified local changes in OB volume over time by measuring distance between individually identified neurons. Indeed, volumetric analysis of inter-neuronal space between groups of four neurons in X, Y, Z (thus an irregular pyramid) demonstrated that distance between neurons increased steadily between 2 and 5 months (*Figures 7a,b* and *4* animals at each time point), strongly indicating continuous OB growth.

Second, we used light sheet microscopy on CUBIC-treated (*Susaki et al., 2014*) brains to investigate the structure and volume of the whole adult mouse OB. Volumetric analysis based on 3D reconstructions of cleared OBs (*Figure 7c,d* and *Video 3*) revealed a steady increase in OB size leading to a significant 44% enlargement of the structure from 2 to 12 months (*Figure 7d,e*). This volume increase affected equally the granule and the glomerular layers (*Figure 7f*) in the absence of obvious changes in layer repartition (*Figure 7—figure supplement 1a*). During the same interval, total forebrain volume was unchanged (*Figure 7g*).

Next, we investigated the evolution of cell density in the more homogeneous granule cell layer using cleared brain tissues. To count all cells in the GCL, we stained nuclei with the fluorescent marker TOPRO3. Quantification revealed that the density of nuclei was highly stable at all observed time points (*Figure 7h,i*) while the density of astrocytes decreased and microglia density was unchanged (*Figure 7—figure supplement 1a, b and c*).

Thus, both in vivo brain imaging and light sheet microscopy of fixed cleared tissue demonstrate that the mouse OB grows significantly during adult life in the absence of detectable changes in cell density. This is in strong support of the permanent addition of new neurons to a stable preexisting circuitry in the absence of substantial cell death.

## Discussion

Our work, combining long-term in vivo observations, pulse chase experiments and 3D morphometric analyses, leads to three main conclusions:

First, the level of neuronal cell death among perinatal and adult born interneurons after their arrival in terminal positions in the OB is very low. Second, adult OB neurogenesis is not a homeostatic but an addition process. Third, classical lineage tracing approaches based on thymidine analogs are associated with unwanted side effects and have to be interpreted with care.

Using a non-invasive long-term imaging approach combined with lineage-tracing approaches using low concentrations of the thymidine analog EdU, we were unable to detect considerable cell loss among postnatal and adult generated neurons during the first weeks after their arrival in the OB.

The predominant evidence that led to the postulation of selection in the OB target layers is based on the use of thymidine analogs, generally BrdU or 3H-dT, that incorporate into the nuclear DNA during the S-phase of the cell cycle (*Petreanu and Alvarez-Buylla, 2002*; *Winner et al., 2002*).

However, both BrdU and 3H-dT are toxic (*Breunig et al., 2007*; *Ehmann et al., 1975*; *Kolb et al., 1999*; *Kuwagata et al., 2007*; *Nowakowski and Hayes, 2000*; *Sekerková et al., 2004*; *Taupin, 2007*) and studies in both rodents (*Lehner et al., 2011*; *Webster et al., 1973*) and primates (*Duque and Rakic, 2011*) pointed to unwanted, and hard to interpret, long term effects associated to their use. Accordingly, warnings concerning the interpretation of such data have been issued (*Costandi, 2011*; *Lehner et al., 2011*).

In agreement with the existing literature, we observed massive loss of newborn neurons in the OB when standard doses of BrdU or EdU were used for tracing. Interestingly, in the presence of considerably lower concentrations of EdU neurogenesis was still obvious but cell loss during the critical period was not detectable anymore. This finding is in perfect agreement with our in vivo observations, in which we find very little neuronal death in the OB layers.

These results lead to the conclusion that a selection step in which an overproduced precursor population is matched to the needs of the target structure, does not principally occur after arrival in

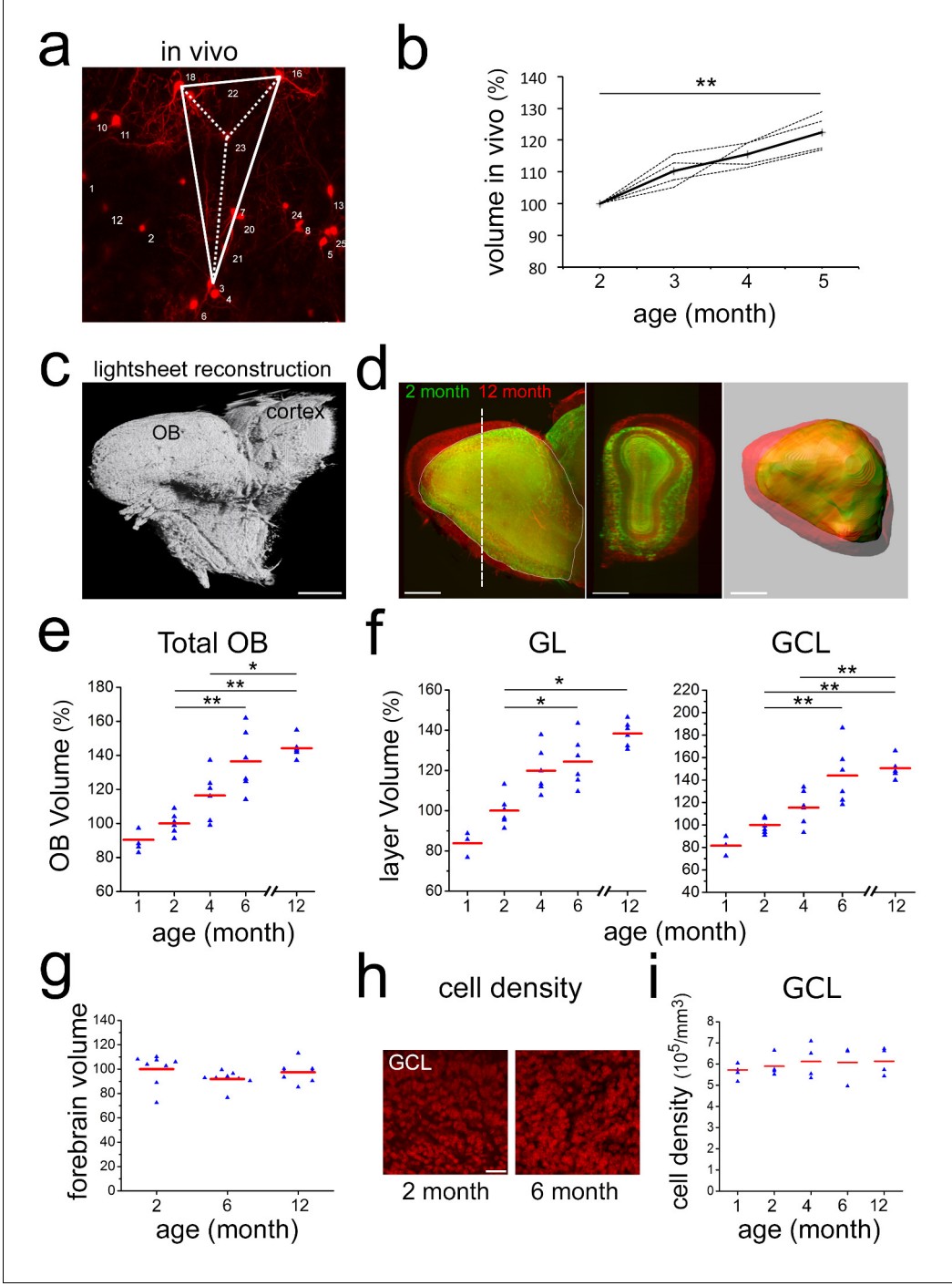

**Figure 7.** Neuronal addition and growth of the OB. (**a**) Example of volumetric analysis of inter-neuronal space between individually identified neurons in vivo (groups of four neurons in X,Y,Z; thus an irregular pyramid). (**b**) Quantification inter-neuronal space shows that distances between identified neurons increase in adult animals. (**c**) Reconstruction of an adult OB and part of the cortex based on 3-D light sheet imaging. (**d**) Comparison of 2- and 12-month-old OBs reconstructed from lightsheet images. (**e**) Quantification of volume increase in OBs from 1 to 12 months, normalized to 2 months. (**f**) Quantification of volume increase in OB sublayers from 1 to 12 months, normalized to 2 months. Volume increase affects both, the GL and the GCL. (**g**) Quantification of the volume of the forebrain at 2, 6 and 12 months, normalized to 2 months. Forebrain size over time does not change. (**h**) Examples of GCL cell density based on TOPRO-3 nuclear staining in whole, cleared OBs of 2- and 6-month-old

*Figure 7 continued on next page*

*Figure 7 continued*
mice. (i) Quantification of cell density from 1 to 12 months. Cell density is constant over the observation period. Scale bar: 800 µm in a,b. 40 µm in f, 50 µm in h.
DOI: https://doi.org/10.7554/eLife.44830.014
The following figure supplement is available for figure 7:

**Figure supplement 1.** Repartition of neurons in the different OB layers over time and density of astrcoyte and microglia over time.
DOI: https://doi.org/10.7554/eLife.44830.015

the OB. While this observation is unexpected, it is not completely isolated. For example, in Bax-KO mice, in which apoptotic cell death is blocked, the general structure and size of the OB neuron layers are virtually unaffected (*Kim et al., 2007*) while a disorganization and accumulation of neuronal precursors in the RMS was observed. This points to the possibility that neuronal selection occurs more at the level of recruitment from the RMS than at the level of integration in the target layers. Such a scenario of 'early selection' is also supported by the observation that the density of apoptotic cells is much higher in the RMS than in the OB proper (*Figure 5i* and *Biebl et al., 2000*). However, other scenarios, like an impact of altered migration on survival in the OB, cannot be excluded.

Alternatively, integration or death of OB interneurons might be intrinsically encoded. It has been shown that in developing cortical interneurons neuronal survival is largely independent of signals from the local environment but that about 40% of the total population is predestined to undergo Bax-dependent apoptosis (*Southwell et al., 2012*). In such a scenario, cell death would be expected to occur already in the SVZ/RMS.

Our results demonstrate that neuronal death is a rare event not only during early stages after arrival in the OB, but at all observed time points. However, the permanent addition of new neurons in the absence of considerable cell removal is not compatible with the idea that the OB represents a turnover system of constant size (*Bergami and Berninger, 2012*; *Imayoshi et al., 2008*). Growth has to be expected and our in vivo imaging and light sheet microscopy studies clearly demonstrate a 40% vol increase during the first year of adulthood in the absence of detectable changes in cell density. Indeed, growth of the adult OB in mice has been observed in other studies, although considerable variation and dependence on genetic background have been reported (*Mirich et al., 2002*; *Richard et al., 2010*). Other studies did not find obvious differences in total OB size or specific sublayers (*Imayoshi et al., 2008*; *Petreanu and Alvarez-Buylla, 2002*; *Pomeroy et al., 1990*). What could be the reason underlying these contradictory findings? Past approaches were based on the 2D analysis of a subset of tissue sections and the extrapolation of the total volume based thereon. However, the OB is not a simple radial symmetric globule, but a complex multi-layered structure that shows huge variations along the rostro-caudal and dorso-ventral axes (see *Video 3*). Light sheet microscopy is suited to overcome many of these limitations as the OB is imaged and measured in its entirety. Extrapolations can be avoided and the selection of comparable levels for layer analyses is simple and reliable. Increases in OB lengths are directly obvious. As a consequence inter-animal variations are minor and growth of the structure becomes evident.

Independently from the light-sheet approach, we show that during in vivo long-term observations constantly larger frames are needed to accommodate the same group of cells and that the distance of individually identified neurons measurably increases. The latter finding is in full

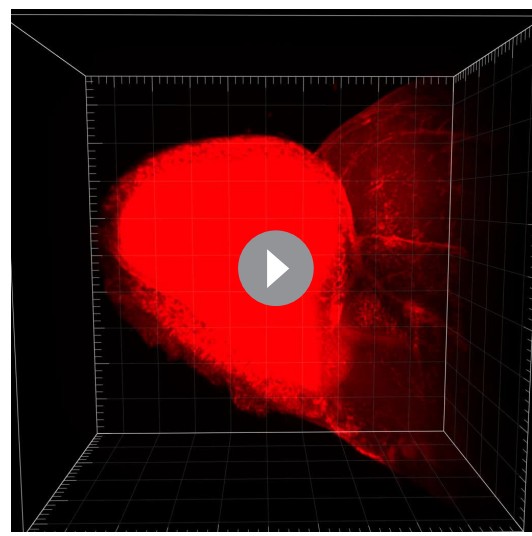

**Video 3.** 3D representation of an adult OB based on light sheet microscopy.
DOI: https://doi.org/10.7554/eLife.44830.016

agreement with the observation that the distance between specific glomeruli increases with age of the animal (*Richard et al., 2010*). Altogether, these data clearly demonstrate growth of the mouse OB during the entire first year of the animal's life.

In conclusion, we show here that neuronal cell death is rare in the OB and that neuronal addition, but not replacement, is the outcome of adult neurogenesis. Genetic fate mapping studies (*Ninkovic et al., 2007*) and also direct observation of the TH positive neuron population (*Adam and Mizrahi, 2011*) already pointed toward an increase in specific neuronal subsets in the adult OB. We show that neuronal addition is a general phenomenon that affects perinatally and adult generated neurons, leading to substantial growth of the OB throughout life. Thus, OB neurogenesis appears to reflect ongoing brain development rather than homeostasis.

How many neurons are added to the adult OB? Our data, based on measurements of cell density and volume of the structure, allows to estimate that between 2 and 6 months about 8000 cells/day are added to the growing OB. Interestingly, using a genetic approach, Imayoshi et al found that 6 months after tamoxifen induction, labeled neurons represent 41.2% of the total population (1,500,000, thus 620,000 new neurons). Considering 60–70% recombination efficiency at the stem cell level (*Imayoshi et al., 2008*), this leads to a number of almost 6000 new cells that are added per day. Thus, our direct measurements and the genetic approach render highly comparable results in terms of the number of neurons that integrates in the adult OB. However, these numbers are substantially lower than estimates based on thymidine analog labeling (*Petreanu and Alvarez-Buylla, 2002*) and future experiments will be needed to address this discrepancy.

Our work leaves of course other open questions. For example, it has been shown that olfactory enrichment and learning increases the survival of newborn neurons in the OB. Neurons are 'saved' from dying apoptotic death (*Mouret et al., 2008*; *Rochefort et al., 2002*; *Sultan et al., 2010*). But how can neurons be saved when death is extremely rare in first place? Repeating in vivo observations and low-dose EdU studies in the context of olfactory stimulation and learning will help to clarify these matters.

Moreover, an increase in BrdU-positive cells in the GCL was observed in BAX conditional mutant mice (*Kim et al., 2007*; *Sahay et al., 2011*). While this could indicate that cell death is a regulating factor in the OB target layers, it is also in agreement with a scenario where neurons are selected during exit from the RMS, as discussed above.

Finally, neuronal selection at the level of integration has been proposed to underlie adult hippocampal neurogenesis (*Bergami and Berninger, 2012*; *Buss et al., 2006*), and recent in vivo observations indicate that most cell death occurs among immature neurons, at relatively early stages after their birth (*Pilz et al., 2018*). This correlates well with the quasi absence of death during later stages, that we observe in the OB.

## Materials and methods

### Animals

All mice were treated according to protocols approved by the French Ethical Committee (#5223–2016042717181477 v2). Mice were group housed in regular cages under standard conditions, with up to five mice per cage on a 12 hr light–dark cycle. 2 months old C57Bl6 males were used for BrdU and edU pulse chase experiments. Rosa-RFP mice (Ai14, Rosa26-CAG-tdTomato; *Madisen et al., 2010*) were obtained from the Jackson laboratory and used on a mixed C57Bl6/CD1 background. For lightsheet experiments, 1, 2, 4,6 and 12 months old male C57Bl6 were obtained from Janvier labs. Nestin-CreERT2 mice were obtained from Amelia Eisch (*Lagace et al., 2007*) and crossed with Rosa-RFP mice. Male and female Nestin-CreER$^{T2}$ X rosa-RFP mice were used between 2- and 3-month old at the time of surgery.

### In vivo labeling of neurons

In vivo electroporation was performed as previously described (*Boutin et al., 2008*). Briefly, 1-day-old pups were anaesthetized by hypothermia and 1 µl of a pCAG-CRE) plasmid (*Platel et al., 2010*) at 4 µg/µl) was injected in the lateral ventricle. Electrical pulses were applied to target the dorsal V-SVZ.

In adult Nestin-CreERT2 X rosa-RFP mice, RFP expression was induced by tamoxifen injection (Sigma-Aldrich; intraperitoneal; dissolved in 10% EtOH/90% sunflower oil) at 100 mg/kg per day for 2 days.

## Surgical preparation

Implantation of an observation window was performed as previously described (*Drew et al., 2010*) but with minor modifications. Briefly, mice were anaesthetized by intraperitoneal (ip.) injection of ketamine/xylazine (125/12,5 mg/kg). Dexaméthasone (0.2 mg/kg) and buprenorphine (0.3 mg/mL) were injected subcutaneously and lidocaine was applied locally onto the skull. The pinch withdrawal reflex was monitored throughout the surgery, and additional anesthesia was applied if needed. Carprofen (5 mg/kg) was injected ip. after the surgery. A steel bar was added during this step to allow fixation of the animal to the microscope. The skull overlying the OB was carefully thinned with a sterile scalpel blade until a thickness of 10–20 µm was reached. A thin layer of cyanoacrylate (superglu3, Loctite) was applied and a 3-mm round coverslip was apposed and sealed with dental cement (superbond, GACD). A first microscopic observation was performed on these anesthetized mice.

For olfactory sensory deprivation, a silicone tube was inserted (Intramedic; 0,5 mm diameter, 3 mm long) into one naris and sealed with cyanoacrylate glue (*Cummings et al., 1997*). Efficiency of occlusion was checked the following day and before each imaging session. At the end of the experiment, immunostaining against tyrosine hydroxylase was performed to confirm the efficiency of occlusion.

## In vivo two-photon imaging

We used a Zeiss LSM 7MP two-photon microscope modified to allow animal positioning under a 20X water immersion objective (1.0 NA, 1.7 mm wd) and coupled to a femtosecond pulsed infrared tunable laser (Mai-Tai, SpectraPhysics). After two-photon excitation, epifluorescence signals were collected and separated by dichroic mirrors and filters on four independent non-descanned detectors (NDD). Images were acquired using an excitation wavelength of 950 nm. RFP was first collected between 605–678. In addition, we collected an additional RFP signal between 560 and 590 that was voluntarily saturated to allow a better identification of subcellular structures like dendrites.

In general, image acquisition lasted about 10 min. Mice could potentially move on a treadmill during imaging, but rarely did so. The imaging window was centered on the dorsal surface of the OB. The whole PG layer was imaged for periglomerular observation experiments (around 150 µm). For GC observation experiments, we imaged from the surface of the olfactory bulb to a depth of 400 to 600 µm.

On consecutive observation, the same field of view was localized based on the geometric motifs of groups of neurons and specific morphological features of individual cells. Between 18 and 179 neurons were imaged initially every week for the first 8 weeks and further imaged at irregular intervals for up to 22 weeks. Images of 606 × 606 µm were acquired at 0.59 µm/pixel resolution in the xy dimension and 2 µm/frame in the z dimension to a maximal depths of 400 µm.

## In vivo imaging of adult born GCs labeled with a lentivirus

See *Wallace et al. (2017)* for cranial window, virus, and imaging. Briefly, 250 nL of undiluted virus (1:1 mixture of lenti-syn-tTAad and lenti-TRE-dTomato-T2A-GCaMP6s) was injected bilaterally at each of two depths to target the RMS (coordinates from bregma: A + 3.3, L ± 0.82, from the brain surface: V-2.9 and −2.7). The virus was locally produced and the viral titer was not measured. Supplementary file 1 in the *Wallace et al. (2017)* shows an example of the injection site in a sagittal slice and demonstrates that the virus does not diffuse into the bulb at the volume and titer we used. All cells in the present analysis were labeled with the tdTomato-GCaMP6s version of the lentivirus.

Analysis: Z stacks taken with a 1 or 2 µm z-step were used for tracking cells over weeks. Maximum intensity projections were created and annotated manually in ImageJ and cross-referenced with z-stacks to confirm that the dendritic structure and location of a cell allowed unambiguous identification. Each line in 3C represents cells tracked for different lengths of time, and multiple lines may correspond to a single field of view. For example, this imaged field of view corresponds to the two lines representing three cells tracked over time, with one line ending at 5 weeks and one ending at 7 weeks (due to the final z stack not extending deep enough to include the first 3 cells). The

newcomers arrived at different times, so they have different lines. For example, some of the cells with asterisks arrived at 3 weeks and others arrived at 4 weeks. Incoming cells not marked with asterisks had cell bodies that either were not fully included within the z stack or we were not able to track them for more than one imaging session and so were not quantified.

## Chronic in vivo imaging analysis

Quantitative analyses were performed on raw image stacks using FIJI software (*Schindelin et al., 2012*). All neurons identified on the first image were assigned a number using ImageJ overlay. Based on morphology and relative position each neuron was individually numbered and tracked on the successive weekly images (see *Figure 1—figure supplement 3b,d*). After identification of the first cohort, smaller numbers of additional neurons appeared permanently in the observation window as a consequence of ongoing neurogenesis in the stem cell compartment (arrowhead in *Figure 1d*, 5wpi). These were numbered and followed like the first cohort. Results were summarized in Microsoft Excel. Occasionally neurons located at the border of an image were placed outside of the imaged field in one of the following sessions. These cells were excluded from further analyses. Animals showing an evident degradation of the imaging window were excluded from further imaging sessions.

For the analyses of the distance between neurons (*Figure 7ab*), we measured over time the volume between four neurons (three neurons in the same plane and another neuron in a different plane) using FIJI. We measured two pyramids (i.e. eight neurons) per animal in four animals from 2 to 5 months.

## Quantification of Brdu and EdU pulse chase experiments

BrdU (Sigma) was injected ip. 4 times at 50 mg/kg body weight every 2 hr. EdU (Sigma) was injected 4 times at 1, 5 or 50 mg/kg body weight every 2 hr. Staining was performed as described previously (*de Chevigny et al., 2012a*). For the dose of 1 mg/kg of edu, the labeling protocol was repeated to increase the intensity of the staining. This was not necessary for the dose of 5 mg/kg. Stainings were done on 50 μm floating vibratome sections. Images were taken either using a fluorescence microscope (Axioplan2, ApoTome system, Zeiss, Germany) or a laser confocal scanning microscope (LSM880, Zeiss, Germany). Conditions were blinded to the experimenter. Labeling at 2 and 6 weeks were performed on the same day. Three to five OB sections were randomly chosen in five animals per condition. Stack of 6 to 15 images were performed every 2 um over the whole surface of the OB slice. The number of BrdU/EdU-positive cells in the glomerular layer and granule cell layer were quantified in three dimension using FIJI software and Imaris software. The results were divided by the volume of the region to give a density of labeled cells per mm3. The mean per animal is represented in *Figure 5* and used for statistical tests.

## Quantification of cleaved-Caspase three immunostaining

We used the same methods as explained in the previous section to measure the density of cleaved caspase3-positive cells. Quantification was performed in 15 animals (5 animals 2 weeks after 5 mg/kg EdU injection, 5 animals 2 weeks after 50 mg/kg EdU injection, 5 animals 6 weeks after 50 mg/kg EdU injection).

## Light-sheet microscopy

To render brains transparent we followed the Cubic protocol (*Susaki et al., 2014*). Briefly, brains where incubated in Cubic1 solution for 10 days at 37°C using gentle agitation. After clearing, brains were incubated for 1 day in the red nuclear dye TOPRO3 (1/1000) in PBS, 0.01% Tween 20, 0.01% sodium Azide at 37°C. The brains were then re-incubated for 3 hr in Cubic1 solution and subsequently placed in Cubic2 solution for 2 days at 37°C. Timing of all steps was carefully monitored.

We used a lightsheet Z1 microscope (Zeiss) with a 5x/0.16NA objective to image the transparized OB and UltraMicroscope II (LaVision BioTec) with LWDO 2x/0.14NA for whole brain imaging. OB layers were easily distinguishable using the nuclear staining of the TOPRO3. The OB was imaged every 5.9 μm in Z with a xy pixel dimension of 2.5 μm and the whole brain with 30 μm steps in Z and a xy resolution of 3.03 μm, respectively. We used Imaris software (Bitplane, Germany) for reconstruction of the total volume based on the nuclear TOPRO3 staining. To determine total forebrain size

we measured the entire volume from the caudal end on the OB to the caudal end of the neocortex. All measurements were normalized to the mean obtained on 2-month-old brains.

## Measurement of cell density

We measured cell density in the granule cell layer by imaging the same transparized brain with the two-photon microscope used for in vivo imaging to obtain a better resolution. We acquired Z stacks of 200 µm with 2 µm resolution in Z and 0.3 µm in xy in the central part of the OB. These images were first de-noised in Fiji using a 3D mean filter. Then the volumetric density of nuclei was quantified using *Imaris* software: We use the cell detection module to detect nuclei in the granule cell layer. We used 4 µm as a seed point value to split the connected objects.

## Immunohistochemistry

Stainings were done on 50 µm floating vibratome sections as described before (*Tiveron et al., 2016*). Primary antibodies: GFP (rabbit IgG, Life technologies,1 1:1000 or chicken Ig, AVES, 1:1000), Calretinin (mouse IgG1, Synaptic Systems; 1:2000), Tyrosine Hydroxylase (chicken Ig, AVES; 1:1000), IBA1 (life technologies, 1:500), GFAP (life technologies, 1:500), cleaved-caspase3 (Cell Signaling Technology: #9662, 1:500). Secondary antibodies were purchased from Life Technologies. Before mounting, cell nuclei were stained with Hoechst 33258 or TOPRO3. Optical images were taken either using a fluorescence microscope (Axioplan2, ApoTome system, Zeiss, Germany) or a laser confocal scanning microscope (LSM880, Zeiss, Germany).

## Statistical analyses

All data are presented as mean ± s.e.m. Statistical comparisons were performed using Matlab software (Mathworks) or R. In box plot representation, center line represents the median; box limits, upper and lower quartiles; whiskers, outliers). All statistical tests were two-tailed. Threshold for significance was set at $p=0.05$. For occlusion experiments (*Figure 2g*) we used a Wilcoxon Rank-sum test ($p=0,0357$). For Brdu and Edu pulse chase experiment (*Figure 3*), we used a Wilcoxon Rank-sum test ($*p<0,05$, $**p<0,01$, $***p<0,001$). For quantification of the volume of the OB and the volume of the layers, we used a wilcoxon ranksum test for each comparison (six comparison) and adjusted the p-value threshold for multiple comparisons using Bonferroni ($*p<0,0083$, $**p<0,00167$, $***p<0,000167$). For the quantification of the increase of size during in vivo imaging experiments (*Figure 5i*), we used a Friedman rank sum test followed with a post hoc test in the Matlab software (between 2 and 5 months $**p<0.01$).

# Acknowledgements

The authors thank Andrea Erni for critical reading of the manuscript, Brice Detailleur for technical help on the two photon microscope. We are particularly grateful to the local PiCSL-FBI core facility (IBDM, AMU-Marseille) supported by the French National Research Agency through the « Investments for the Future' program (France-BioImaging, ANR-10-INBS-04) as well as the IBDM animal facilities. We thank Francois Michel from INMED for help with light sheet microscopy. Work in the Murthy lab on neurogenesis was supported by NIH grant R01DC013329, and JLW was supported by NIH F31DC016482. This work was supported by Agence National pour la Recherche (grant ANR- 13-BSV4-0013), Fondation pour la Recherche Médicale (FRM) grants ING20150532361, FDT20160435597 to HC and FTD20170437248 to AA, Fondation de France (FDF) grant FDF70959 to HC.

# Additional information

## Funding

| Funder | Grant reference number | Author |
|---|---|---|
| Agence Nationale de la Recherche | ANR-13-BSV4-0013 | Harold Cremer |
| Fondation pour la Recherche Médicale | ING20150532361 | Harold Cremer |

| Fondation pour la Recherche Médicale | FDT20160435597 | Harold Cremer |
| Fondation pour la Recherche Médicale | FTD20170437248 | Alexandra Angelova |
| Fondation de France | FDF70959 | Harold Cremer |
| National Institutes of Health | R01DC013329 | Venkatesh N Murthy |
| National Institutes of Health | F31DC016482 | Jenelle Wallace |

The funders had no role in study design, data collection and interpretation, or the decision to submit the work for publication.

### Author contributions

Jean-Claude Platel, Conceptualization, Data curation, Formal analysis, Supervision, Funding acquisition, Validation, Investigation, Methodology, Writing—original draft, Writing—review and editing; Alexandra Angelova, Stephane Bugeon, Jenelle Wallace, Thibault Ganay, Ilona Chudotvorova, Jean-Christophe Deloulme, Christophe Béclin, Marie-Catherine Tiveron, Nathalie Coré, Investigation; Venkatesh N Murthy, Supervision, Investigation; Harold Cremer, Conceptualization, Supervision, Funding acquisition, Writing—original draft, Project administration, Writing—review and editing

### Author ORCIDs

Jean-Claude Platel https://orcid.org/0000-0001-5542-3076
Venkatesh N Murthy http://orcid.org/0000-0003-2443-4252
Harold Cremer https://orcid.org/0000-0002-8673-5176

### Ethics

Animal experimentation: All mice were treated according to protocols approved by the French Ethical Committee (#5223-2016042717181477v2).

### Decision letter and Author response

Decision letter https://doi.org/10.7554/eLife.44830.020
Author response https://doi.org/10.7554/eLife.44830.021

## Additional files

### Supplementary files

• Source data 1. Source data.
DOI: https://doi.org/10.7554/eLife.44830.017
• Transparent reporting form
DOI: https://doi.org/10.7554/eLife.44830.018

### Data availability

The raw data that support the findings of this study are several TBs in size and are therefore available on request. A source data file for the main figures has been provided.

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
