## [Decision Letter]

[Editors’ note: this article was originally rejected after discussions between the reviewers, but the authors were invited to resubmit after an appeal against the decision.]

Thank you for submitting your work entitled "Neuronal integration in the adult olfactory bulb is a non-selective addition process" for consideration by *eLife*. Your article has been reviewed by three peer reviewers, and the evaluation has been overseen by a Reviewing Editor and a Senior Editor. The following individuals involved in review of your submission have agreed to reveal their identity: Richard Nowakowski (Reviewer #2).

Our decision has been reached after consultation between the reviewers. Based on these discussions and the individual reviews below, we regret to inform you that your work will not be considered further for publication in *eLife*.

Summary:

Overall, this is a potentially interesting manuscript that highlights how experimental differences may lead to very different outcomes/interpretations between studies. As you will see below, there was enthusiasm for this topic and the approach. However, a main concern of the reviewers in the post-review discussion was the strong statements, intended to counter many previous studies. This in itself is not bad, but the experimental rigor is lacking to fully support claims of a strong paradigm shift. In general, that would require consideration of additional data, other experimental approaches, and/or larger sample sizes. At present, the conclusions are over-interpreted. The dose-dependent effects of thymidine analogs suggest that high doses kill cells, but whether the low dose is sufficient to capture a decrease in number due to cell death if less clear. The number of neurons imaged with 2P seems too small to make a definitive conclusion about cell-death.

*Reviewer #1:*

This is an interesting study that challenges the notion that integration of adult-granule cells is dependent on neuronal competition. The authors show that normally used doses of thymidine analogs induce cell-death whereas low doses fail to do so. Using 2P imaging, the authors do not find evidence of cell death of adult-born granule cells "during a critical period" up till 8 weeks.

- The sample size for imaging is on the smaller side: 48 neurons from 3 mice (from 1-7 weeks post injection). Can the authors make conclusions regarding activity dependent survival based on such a small number of cells?

- It is not clear why the authors did not use the Nestin CreERT2:Ai14 system to label adult-born granule cells for 2P imaging. Is Lentiviral labeling an equivalent approach (given the effects of injury, inflammation etc.)?

- Conditional deletion of Bax in nestin+ cells in the adult SVZ results in increased survival of adult-born granule cells (Sahay et al., 2011). Are the authors suggesting that Bax dependent cell-death plays a role prior to neuronal integration?

- Is it possible that activity dependent competition plays a role at a later stage?

- Neuronal competition is suggested to guide integration of adult-born dentate granule cells (Tashiro Nature 2006, Adlaf et al. *ELife* 2017, McAvoy et al., Neuron 2016). Do the authors think that competition occurs only in the DG niche?

*Reviewer #2:*

This is a great paper! It has impact on the OB issues and also on the DG related proliferation. It is difficult to argue with the single cell data as it has significant advantages over the generally used S-phase labeling.

The missing aspects of this paper are the early time points and a quantitative analysis (i.e., model). The first point should be addressed in the paper regarding what limits the missing early time points place on the conclusions – there are a few. The quantitative analysis, i.e., model could be the topic of a follow up manuscript.

*Reviewer #3:*

In the manuscript by Platel et al., the authors present an intriguing argument against neuronal turnover in the OB, and against critical periods for adult born neuron survival. They show that (1) populations of OB neurons theoretically replaced by adult neurogenesis are actually quite stable, (2) common methods for labeling adult born neurons via intercalating agents can lead to cell death, and (3) the volume of the OB increases continuously with age in accordance with a net influx of neurons. By implementing chronic two photon imaging of genetically labeled neurons, the authors show that populations of perinatally-born and adult-born PGCs are much more stable than previously appreciated. Whereas previous studies showed up to 50% loss of newborn neurons after 4 wks, this study shows little to no reduction in cell numbers. The authors pose that this is mainly due to differences in labeling strategy, and present intriguing evidence that labeling developing neurons with high concentrations of BrdU/EdU induces artificially high cell death, and that when low concentrations of EdU are used there is limited- or no loss of EdU-labeled cells over time. From these findings, the authors make the argument that adult neurogenesis in the OB is not a homeostatic process where the cell loss is balanced by the addition of new neurons, and that the influx of cells corresponds to a continual increase in OB volume. This finding is surprising and a bit harder to reconcile with the notion that the adult brain is in a relatively steady state. They use a combination of extrapolation within fields of view in vivo and whole OB imaging at different ages to show that the volume increases, but cell density and architecture within and between layers remain the same. Overall, this is a useful and informative study that should certainly be considered at *eLife*, however key aspects of the manuscript should definitely be addressed prior to acceptance for publication.

Concerns:

• Although provocative to make the main claim that adult born neurons do not show a "critical period", this overall notion is still challenged by the fact that survival/integration is still directly influenced by sensory deprivation/enrichment. This is a key idea to critical period…that levels of activity influence neuronal development. Until this is more clearly fleshed out with further enrichment/learning/training paradigms, it remains a stretch to turn over predominant models, and thus the language should be tempered.

• The in vivo imaging data is appreciated, and certainly reveals very important discrepancies between previous studies and the current work regarding turnover rates in adult born neuron populations. However, I find it a bit surprising that the authors did not present in vivo imaging data from animals actually labeled with differing concentrations of EdUA. Performing similar analyses in the variable-dosed animals and observing differential turnover rates in vivo would be very compelling, since it would be a direct comparison rather than completely separate experiments. This is a key point, and if not directly addressed with further experimentation, must at least be discussed.

• The study is lacking details in the Materials and methods section regarding viral injections and imaging methods. From the reference, it is unclear what total volume, titer, and dilution of virus was injected. This is important to assess aspects of labeling progenitor vs. migrating/postmitotic populations through potential diffusion, etc. On the same note, the authors do not describe how they chose their fields of view for imaging, including orientation, depth relative to the surface of the bulb, or other layer landmarks in the OB. Also, each field of view obtained from the same animal should not be considered an independent sample. Especially with only 3 animals, they should account for this by using a nested analysis or a linear mixed model. As is, the current Materials and methods section makes it difficult to fully evaluate this experiment.

• The description of overall bulb growth using triangulation measurements is appreciated, especially with reference to other brain areas. But given the estimates of newborn neuron birth rates, and the posed limited turnover, it would be possible to generate models and/or estimates of how much the OB should expand. The described growth rate here, still seems quite small considering the robust rate of addition vs. cell death. Can this be modeled, measured, and/or reconciled?

• For light sheet imaging, the authors use CUBIC-treated samples. Many of the recently developed clearing techniques impact final tissue volumes. It is important to show that this method is not affecting the brain volume differentially at different ages and different structures. Alternatively, other methods such as small animal in vivo CT scans that offer direct volume measurements might be considered.

• Finally, the authors do not present any data or discussion of how other OB cell types that do not undergo adult neurogenesis compensate for the increase in volume and/or presumed impact on OB circuitry. These issues should at least be raised in the Discussion.

---

## [Author Response]

[Editors’ note: the author responses to the first round of peer review follow.]

Thank you for reviewing our work. Based on the reviews provided, we are very surprised by the decision to reject the paper. Reviewer 1 mentions,”This is an interesting study that challenges the notion that integration of adult-granule cells is dependent on neuronal competition”. Reviewer 2, “This is a great paper!”. Reviewer 3, “the authors present an intriguing argument against neuronal turnover in the OB, and against critical periods for adult born neuron survival.” “Overall, this is a useful and informative study that should certainly be considered at e*Life*”.

However, we also understand that during the post-review discussion this initially positive picture changed and that you feel that our strong statements contradicting previous findings are not backed up by sufficiently strong data. In this letter we would like to provide some additional information concerning the wording in our manuscript and the data that we provide.

Regarding your point concerning the "… the strong statements, intended to counter many previous studies." After having received your rejection we went back to the manuscript. We agree that some of the statements are quite strong and should be toned down. However, please consider that this work has been reviewed and revised before, and some of the strong statements are a consequence of demands by previous reviewers. The initial version was much 'softer'. We agree that we should go back to that.

Reviewer #1:

[…] - The sample size for imaging is on the smaller side: 48 neurons from 3 mice (from 1-7 weeks post injection). Can the authors make conclusions regarding activity dependent survival based on such a small number of cells?- It is not clear why the authors did not use the Nestin CreERT2:Ai14 system to label adult-born granule cells for 2P imaging. Is Lentiviral labeling an equivalent approach (given the effects of injury, inflammation etc.)?

To address this point we repeated the entire experiment of observing adult born GCs over several months. As suggested by the reviewer, we induced cohorts of newborn neurons using the Nestin CreERT2:Ai14 system and followed granule cells over 8 weeks and beyond, up to 20 weeks. Using a chronic window approach we were able to follow 101 cells at high resolution. The results are in perfect agreement with the data for all cell populations observed, regardless if electroporation, viral transduction or Nestin-CRE-ERT2 was used: Once cells appear in the OB target layers, loss is very rarely observed. This does not exclude that a selection process based on recruitment occurs at an earlier time point, as discussed below.

Both datasets are now presented in Figure 3. Thus, for adult born granule cells, analyses based on different labeling approaches and performed in different labs led to the same conclusion.

- Conditional deletion of Bax in nestin+ cells in the adult SVZ results in increased survival of adult-born granule cells (Sahay et al., 2011). Are the authors suggesting that Bax dependent cell-death plays a role prior to neuronal integration?

Yes, that is indeed what we think. We say in the Discussion "… in Bax-KO mice, in which apoptotic cell death is blocked, the general structure and size of the OB neuron layers are virtually unaffected (Kim et al., 2007) while a disorganization and accumulation of neuronal precursors in the RMS was observed. This points to the possibility that neuronal selection occurs more at the level of recruitment from the RMS than at the level of integration in the target layers. Such a scenario of "early selection" is also supported by the observation that the density of apoptotic cells is much higher in the RMS than in the OB proper (Figure 5I and Biebl et al., 2000).

However, both Sahay et al. and Kim et al. found an increase in BrdU labeled cells in the OB in these mutants. To address this point we say now in the Discussion of our manuscript: "Moreover, an increase in BrdU positive cells in the GCL was observed in BAX conditional mutant mice (Kim et al., 2007; Sahay et al., 2011). While this could indicate that cell death is a regulating factor in the OB target layers, it is also in agreement with a scenario where neurons are selected during exit from the RMS…". We hope this answers the reviewer’s question.

- Is it possible that activity dependent competition plays a role at a later stage?- Neuronal competition is suggested to guide integration of adult-born dentate granule cells (Tashiro Nature 2006, Adlaf et al. ELife 2017, McAvoy et al., Neuron 2016). Do the authors think that competition occurs only in the DG niche?

We do not have evidence for competition impacting on survival at later stages in the OB. All population that we observed for prolonged time windows (up to 6 months) show a high degree of stability and very little neuron loss. Only when we challenge the system using complete odor deprivation we see cell death of mature neurons (Figure 4 but see also our recent paper by Angelova et al., 2019).

As mentioned above, we believe that in the OB neurogenic system competition occurs before arrival at their final destination. We agree that the references cited by the reviewer strongly indicate that synaptic competition occurs in the hippocampus. However, the knowledge is far more fragmentary in the OB.

Reviewer #2:

[…] The missing aspects of this paper are the early time points and a quantitative analysis (i.e., model). The first point should be addressed in the paper regarding what limits the missing early time points place on the conclusions – there are a few. The quantitative analysis, i.e., model could be the topic of a follow up manuscript.

We see the reviewers point. As we cannot image deep structures like the SVZ or the RMS we can only conclude on survival and death at once the new neurons arrived in their terminal positions in the OB. We agree that this was insufficiently clear in the original version. Now we say throughout the manuscript that our observations are valid only for time points "… after arrival in terminal positions in the OB." In addition, we discuss that if selection occurs, this would likely occur in the RMS, which we cannot directly observe, limiting the conclusions. We agree that an extensive quantitative analysis and modeling approaches would be highly interesting. We started collaborations to this aim.

Reviewer #3:

[…] Concerns:• Although provocative to make the main claim that adult born neurons do not show a "critical period", this overall notion is still challenged by the fact that survival/integration is still directly influenced by sensory deprivation/enrichment. This is a key idea to critical period…that levels of activity influence neuronal development. Until this is more clearly fleshed out with further enrichment/learning/training paradigms, it remains a stretch to turn over predominant models, and thus the language should be tempered.

We see the reviewers point and tempered our conclusions wherever this appeared appropriate. In particular, we were cautious when using the term "critical period' as this term is also used for the critical time window of synaptic plasticity that is associated to neuronal integration. We only look at survival after neurons arrived in terminal positions, and this was insufficiently clear in the initial version of the manuscript.

• The in vivo imaging data is appreciated, and certainly reveals very important discrepancies between previous studies and the current work regarding turnover rates in adult born neuron populations. However, I find it a bit surprising that the authors did not present in vivo imaging data from animals actually labeled with differing concentrations of EdUA. Performing similar analyses in the variable-dosed animals and observing differential turnover rates in vivo would be very compelling, since it would be a direct comparison rather than completely separate experiments. This is a key point, and if not directly addressed with further experimentation, must at least be discussed.

This is an important point raised by the reviewer. Indeed, we performed such an experiment: following individual neurons in the OB in the presence of BrdU. When we used 50 mg/kg BrdU we saw a slight increase in cell death during imaging, but did not reach significance levels at p<005. When we use 300 mg/kg we observed significant death levels, however, well below 40%. The data is shown in Author response image 1 and we can introduce it in the manuscript, but we are not sure it is sufficiently strong and conclusive.

The reason underlying this phenomenon is like the following:

- Using nestin-CRE-ERT2 we express CRE in stem cells, type C cells and even some type A cells. All recombined cells will express equivalent levels of RFP once arrived in the OB and represent the population that we observe.

However, using thymidine analogue injections the situation is quite different:

- Some type C/type A cells will perform their last S-phase in the presence of BrdU/EdU and incorporate high levels of the analogues. These are likely prone to die in the OB.

- Many cells will not be in S-phase while BrdU is bio-available but will nevertheless induce CRE dependent RFP expression. These will likely live.

- Many cells in the SVZ will be in early phases of their amplification cycle and will dilute their BrdU/EdU content during subsequent divisions. As toxicity of thymidine analogues is dose dependent, cells that underwent several divisions will be less prone to die.

As a consequence only few RFP positive neurons on the OB will contain high amounts of BrdU/EdU in their DNA. This is probably why we need considerably higher doses of BrdU (4x300 mg/kg) to observe significantly increased death by in vivo imaging.

This interpretation is supported by a control experiment that we performed: Co-injection of tamoxifen in adult mice to induce RFP expression combined with BrdU injection at 4x50 mg/kg. Author response image 2 shows an example of cells in the glomerular layer two weeks later. Only one among the different RFP (red) and BrdU positive cells (green) is double positive. Thus, while all approaches that we used, Nestin-CRE-ERT2, CRE electroporation in neonates, viral transduction in the RMS and thymidine analogues label young neurons, we are far away from saturation and significant overlap. Even when we introduced various delays between tamoxifen and BrdU injection we never saw higher overlaps.

**Author response image 2. respfig2:** 

• The study is lacking details in the Materials and methods section regarding viral injections and imaging methods. From the reference, it is unclear what total volume, titer, and dilution of virus was injected. This is important to assess aspects of labeling progenitor vs. migrating/postmitotic populations through potential diffusion, etc. On the same note, the authors do not describe how they chose their fields of view for imaging, including orientation, depth relative to the surface of the bulb, or other layer landmarks in the OB. Also, each field of view obtained from the same animal should not be considered an independent sample. Especially with only 3 animals, they should account for this by using a nested analysis or a linear mixed model. As is, the current Materials and methods section makes it difficult to fully evaluate this experiment.

We repeated this entire experiment of observing adult born GC. The new approach is in detail described in Materials and methods. concerning the virus injection that is still in the paper, we added more information. In particular we say that: "250 nL of undiluted virus (1:1 mixture of lenti-syn-tTAad and lenti-TRE-dTomato-T2A-GCaMP6s) was injected bilaterally at each of two depths to target the RMS (coordinates from bregma: A+3.3, L+/-0.82, from the brain surface: V-2.9 and -2.7). […] Supplementary Figure 1 in the Wallace et al. paper shows an example of the injection site in a sagittal slice and demonstrates that the virus does not diffuse into the bulb at the volume and titer we used."

Finally, the description of the selected field of view is now added to the Materials and methods section: “The imaging window was centered on the dorsal surface of the OB. The whole PG layer was imaged for periglomerular observation experiments (around 150 μm). For GC observation experiments we imaged from the surface of the olfactory bulb to a depth of 400 to 600 μm.”

• The description of overall bulb growth using triangulation measurements is appreciated, especially with reference to other brain areas. But given the estimates of newborn neuron birth rates, and the posed limited turnover, it would be possible to generate models and/or estimates of how much the OB should expand. The described growth rate here, still seems quite small considering the robust rate of addition vs. cell death. Can this be modeled, measured, and/or reconciled?

We addressed this point in the initial manuscript and present a rough calculation in the Discussion. Indeed, our estimation is very close from the ones obtained with genetic approaches (Imayoshi et al., 2008). We say:

“How many neurons are added to the adult OB? Our data, based on measurements of cell density and volume of the structure, allows to estimate that between 2 and 6 months about 8000 cells/day are added to the growing OB. […] However, these numbers are substantially lower than estimates based on thymidine analogue labeling (Petreanu and Alvarez-Buylla, 2002) and future experiments will be needed to address this discrepancy.”

We hope this addresses the reviewer’s concern.

• For light sheet imaging, the authors use CUBIC-treated samples. Many of the recently developed clearing techniques impact final tissue volumes. It is important to show that this method is not affecting the brain volume differentially at different ages and different structures. Alternatively, other methods such as small animal in vivo CT scans that offer direct volume measurements might be considered.

To address this point we performed a control experiment. We measured on the same brains that were used for OB size calculation the volume of the entire forebrain. While the OB showed significant growth the forebrain was stable in size. This data is presented in Figure 7G.